# The NAC Protein from *Tamarix hispida*, *ThNAC7*, Confers Salt and Osmotic Stress Tolerance by Increasing Reactive Oxygen Species Scavenging Capability

**DOI:** 10.3390/plants8070221

**Published:** 2019-07-12

**Authors:** Zihang He, Ziyi Li, Huijun Lu, Lin Huo, Zhibo Wang, Yucheng Wang, Xiaoyu Ji

**Affiliations:** 1State Key Laboratory of Tree Genetics and Breeding (Northeast Forestry University), Harbin 150040, China; 2Key Laboratory of Biogeography and Bioresource in Arid Land, Xinjiang Institute of Ecology and Geography, Chinese Academy of Sciences, Urumqi 830011, China

**Keywords:** NAC transcription factor, *Tamarix hispida*, abiotic stress, ROS scavenging, transcription activation, transient expression

## Abstract

Plant specific NAC (NAM, ATAF1/2 and CUC2) transcription factors (TFs) play important roles in response to abiotic stress. In this study, we identified and characterized a NAC protein, ThNAC7, from *Tamarix hispida*. *ThNAC7* is a nuclear localized protein and has transcriptional activation activity. *ThNAC7* expression was markedly induced by salt and osmotic stresses. Transiently transformed *T. hispida* seedlings overexpressing *ThNAC7* (OE) or with RNA interference (RNAi) silenced *ThNAC7* were generated to investigate abiotic stress tolerance via the gain- and loss- of function. Overexpressing *ThNAC7* showed an increased reactive oxygen species (ROS) scavenging capabilities and proline content, which was accomplished by enhancing the activities of superoxide dismutase (SOD) and peroxidase (POD) in transiently transformed *T. hispida* and stably transformed *Arabidopsis* plants. Additionally, *ThNAC7* activated these physiological changes by regulating the transcription level of *P5CS*, *SOD* and *POD*
*genes*. RNA-sequencing (RNA-seq) comparison between wild-type and *ThNAC7*-transformed *Arabidopsis* showed that more than 40 known salt tolerance genes might regulated by *ThNAC7*, including stress tolerance-related genes and TF genes. The results indicated that *ThNAC7* induces the transcription level of genes associated with stress tolerance to enhance salt and osmotic stress tolerance via an increase in osmotic potential and enhanced ROS scavenging.

## 1. Introduction

Environmental constraints, including cold, salt and drought, which negatively affect the growth and development of plants, are leading large agricultural production losses worldwide [1]. Plants have evolved a wide range of adaptations to abiotic and biotic stresses, involving stress perception coupled with signal transduction and amplification [2]. In progress, TFs can regulate the expression of downstream functional genes involved in biotic or abiotic stress tolerance [3]. Recently, many researches have reported that a battery of TFs or functional genes involved in abiotic stress responses. For instance, overexpression of *AtbHLH112* increased salt and drought tolerance via increasing the expression level of *PODs* and *SOD*s to improve ROS scavenging ability in *Arabidopsis* [4]. Overexpression of *BplMYB46* in transgenic *Betula platyphylla* improved abiotic tolerance via enhancing the ROS scavenging and proline level [5]. Overexpressing of *VvWRKY30* improved salinity stress resistance via regulating osmoticum accumulation and ROS scavenging in Arabidopsis [6]. Besides, the overexpression of *OsDhn1* increased drought and salt tolerance via reducing ROS accumulation in transgenic rice [7]. The overexpression of *AnnSp2* enhanced salt and drought tolerance by ABA synthesis and the elimination of ROS in transgenic tomato [8].

The NAC TF family is a plant specific TF family and includes 117 members in *Arabidopsis thaliana*, 163 in *Populus trichocarpa*, 151 in *Oryza sativa*, and 152 in both *Nicotiana tabacum* and *Glycine max* [9]. NAC proteins possess a nuclear localization signal sequence, which have variable C-terminal domain in addition to highly conserved N-terminal binding domain [10].

In recent years, many studies have reported that a number of NAC TFs involved in biotic [11,12] and abiotic [12,13] stress responses in plants. A relatively large proportion of rice and *Arabidopsis* NAC TFs under different abiotic and biotic stresses showed differential expression patterns by transcriptional analysis. [14,15]. Moreover, NAC TFs confer abiotic stress tolerance to transgenic plants. For instance, *AtNAC72*, *AtNAC109*, and *AtNAC55* contribute to drought tolerance by promoting the detoxification of aldehydes in the glyoxalase pathway [16,17]. Transgenic rice plants overexpressing *OsNAC2*, *OsNAC3*, *OsNAC6*, or *OsNAC14* enhanced drought tolerance [18,19,20,21]. The overexpression of *OsNAC6* induced tolerance to high salt, drought and blast disease in rice [22]. Overexpression of *TaNAC69* improved the expression of stress-related genes and drought tolerance in transgenic wheat plants [23]. Overexpression of *TsNAC1* in both *Thellungiella halophila* and *Arabidopsis* could enhance abiotic stress resistance, particularly salt stress tolerance [24]. *MusaNAC68* positively regulates salt and drought stress tolerance in transgenic banana plants [25]. Overexpression of *MdNAC1* enhanced drought stress tolerance via enhancing photosynthesis and ROS scavenging in apple transgenic plants [26]. Although a variety of NAC proteins in different plant species have been identified and reported to play key roles in abiotic stress, few studies have been performed about the biological functions of NAC TFs in halophytic woody plants.

*Tamarix hispida*, belong to the genus *Tamarix*, is a woody halophyte that it can grow better than other plants in drought and saline soils. *T. hispida* has properly effective stress defense system and is appropriately used to study abiotic stress tolerance. Previous study identified 16 NAC TFs in transcriptomes of *T. hispida* [27]. Among these NACs, we further identified a NAC gene, *ThNAC7*, which can respond to high levels of salt stress. Therefore, we selected *ThNAC7* for further study. The results demonstrated that *ThNAC7* regulates osmotic and salt stress tolerance via modulating the transcription levels of genes that may activate a number of physiological variations, containing ROS scavenging and changes in osmotic potential, in both transiently transformed *T. hispida* and stably transformed *Arabidopsis* plants. The present study offers new insights into the functions of NAC TFs in abiotic stress tolerance.

## 2. Results

### 2.1. Bioinformatics Analysis of ThNAC7

The open reading frame (ORF) of *ThNAC7* is 1737 bp, encodes 578 amino acids with 65.13 kDa molecular weight. Multiple sequence alignment analysis indicated that ThNAC7 and NAC proteins of other plant species shared highly conserved binding domain at the N-terminal. ThNAC7 contains a classic NAC domain that is the basic characteristic of the NAC family (Appendix A). Phylogenetic tree was constructed in order to study the evolutionary relationships between ThNAC7 and other 105 NACs from *Arabidopsis*. The results showed that ThNAC7 is most similar to two *Arabidopsis* NAC proteins, ANAC091 (AT5G24590) and ANAC062 (AT3G49530) from NAC subfamily VI (Figure 1), indicating that ThNAC7 should also belong to the NAC subfamily VI.

ThNAC7 and 88 NACs from the transcriptomes of *T. hispida* were used for another phylogenetic tree to study the phylogenetic relationships of the NAC TF family in *T. hispida*. All NACs could be divided into 10 subfamilies (I–X), and ThNAC7 was most similar to ThNAC35, and belongs to NAC subfamily I (Appendix A).

### 2.2. ThNAC7 Is a Nuclear Localization Protein with Transactivation Activity

The 35S::ThNAC7-GFP (green fluorescent protein) or 35S::GFP plasmid was respectively imported into onion epidermal cells via particle bombardment. Microscopic observation indicated that the 35S::ThNAC7-GFP fusion protein was observed exclusively in nucleus, while the 35S::GFP protein in the control was uniformly distributed throughout the cytoplasm and nucleus (Figure 2A). The results indicate d that ThNAC7 is a nuclear localization protein.

To confirm the transactivation activity of ThNAC7, full length or truncated CDSs of *ThNAC7* were constructed to pGBKT7 vector and transformed into Y2HGold cells to detect their transactivation activity (Figure 2B). All transformants could grow normally on the SD/-Trp plates, showing that they had been transformed in Y2HGold cells. The yeast cells containing the full-length CDS of ThNAC7 grew well and appeared blue on the SD/-Trp-His/X-a-Gal plates, suggesting that this protein is capable of transcriptional activation. Furthermore, we noticed that the yeast cells transformed with the assumed C-terminal activation domain (dc2, amino acids 146–578) of ThNAC7 thrived and became blue, whereas cells expressing the N-terminal domain (dc1, amino acid 1–435) of ThNAC7 could not grow, showing that the C-terminal of ThNAC7 has transactivation activity. The truncated CDSs expressing amino acids 146–290 and 291–435 retained the transactivation activity, whereas cells expressing the 436–578 fragment had completely abolished transactivation activity. Moreover, the smallest fragment of ThNAC7 that retained transactivation activity was located respectively in amino acids 216–290, 291–360, and 361–435. Taken together, the results indicated that ThNAC7 has transcriptional activation with three independent activation domains: aa 216–290, 291–360 and 361–435, respectively.

### 2.3. The Expression of ThNAC7 Is Induced by Osmotic and Salt Stress

QRT-PCR assay proved that the expression of *ThNAC7* was markedly induced by osmotic (400 mM mannitol) or salt (300 mM NaCl) stress in both roots and leaves of *T. hispida* (Figure 3). Under salt stress conditions, *ThNAC7* was highly induced at 12 h in root tissue; meanwhile, it was highly induced at 24 h in leaf tissue (Figure 3A,B). Under osmotic stress conditions, *ThNAC7* was highly induced at 12 h in leaf tissue and at 6 h in root tissue (Figure 3C,D). These results showed that the expression of *ThNAC7* is induced by osmotic and salt stress, implying that *ThNAC7* plays roles in osmotic and salt stress responses.

### 2.4. Overexpression of ThNAC7 in Arabidopsis Confers Osmotic and Salt Stress Tolerance

Nine independent T_3_ homozygous lines overexpressing *ThNAC7* were obatined to verify the molecular function of *ThNAC7* in salt and osmotic stress tolerance. RT-PCR and qRT-PCR results showed that the transgenic *Arabidopsis* successfully expressed the exogenous *ThNAC7* (Appendix A). Moreover, the transcription of *ThNAC7* in OE2 and OE5 lines was higher than that in other lines, so these two lines were chosen for further study (Appendix A). In normal conditions, the OE2 and OE5 had no substantial distinction in growth phenotype compared to that of the Col-0 plants (control) (Figure 4A–C). However, OE2 and OE5 showed greatly enhanced root growth and fresh weight in osmotic or salt stress compared with those in the Col-0 (Figure 4A–C). Col-0 and transgenic *Arabidopsis* plants had approximately the same chlorophyll contents before treatment, whereas the OE2 and OE5 lines both had markedly higher chlorophyll contents than Col-0 plants in salt or osmotic stress (Figure 4D). In addition, phenotypic growth and fresh weight analyses of the plants growing in soil also indicated that OE2 and OE5 grew much better and had higher fresh weights than Col-0 plants (Figure 4E,F). The results suggested that *ThNAC7* plays positive regulatory roles in the regulation of salt and osmotic stress.

### 2.5. Transient Overexpression or Knockdown of ThNAC7 in T. hispida Plants

To further study the role of *ThNAC7* using gain- and loss- of function studies, we confirmed the expression level of *ThNAC7* in overexpressing *ThNAC7* (OE), RNAi-silenced *ThNAC7* (IE), and control plants (transfected with pROKII) using qRT-PCR. We used the transcription level of *ThNAC7* in plants transformed for 48 hours in control plants as the standard (designated as 1) to standardize the corresponding expression of *ThNAC7* in the other *T. hispida* seedlings. The OE plants showed markedly enhanced levels of *ThNAC7* transcripts, while they were dramatically reduced in the IE plants in comparison with those in the control *T. hispida* plants (Appendix A), which indicated that *ThNAC7* had been successfully overexpressed or knocked down; therefore, these transgenic plants were suitable to study the gain- and loss-of-function of *ThNAC7*.

### 2.6. The ROS Scavenging Capability Was Improved by Overexpressing ThNAC7

Reactive oxygen species (ROS) accumulation is a substantial indicator to detect the degree of stress tolerance [26]. To determine ROS accumulation, two major ROS species, superoxide anion (O_2_^−^) and hydrogen peroxide (H_2_O_2_) were stained using nitroblue tetrazolium (NBT) and diaminobenzidine (DAB) in situ, respectively. Under salt and osmotic stress conditions, both NBT and DAB histochemical staining of *T. hispida* plants indicated that the OE *T. hispida* plants had the lowest levels of O_2_^−^ and H_2_O_2_; while the IE had the highest levels (Figure 5A,B). Consistent with the DAB staining results, the determined content of H_2_O_2_ in the OE *T. hispida* plants was the lowest; and the IE *T. hispida* plants showed the highest H_2_O_2_ content in salt and osmotic stress environment (Figure 5C). The above analyses were repeated in *Arabidopsis*. Under salt or osmotic stress environment, NBT and DAB histochemical staining and H_2_O_2_ content assay verified that ROS contents were markedly decreased in OE lines rather than in Col-0 plants (Appendix A). SOD and POD activities, the two primary ROS scavenging enzymes, were further researched in *T. hispida* plants. In normal growth conditions, the POD and SOD activities of OE plants were not significantly different from those of IE and control plants (Figure 5D,E). However, under salt or osmotic stress environment, markedly higher activities of POD and SOD were detected in OE plants; while the IE plants had greatly lower POD and SOD activities compared to those in the control plants (Figure 5D,E). These analyses were repeated in *Arabidopsis* plants. The SOD and POD activity analysis showed that the OE plants had markedly enhanced the SOD and POD activities compared to those of the Col-0 *Arabidopsis* plants in osmotic and salt stress environment (Appendix A). These results showed that *ThNAC7* expression results in reduced ROS accumulation in plant cells, mediated by increased POD and SOD activities.

### 2.7. The Expression Levels of SODs and PODs Are Induced by ThNAC7

We further analysed the transcription levels of *SODs* and *PODs* to confirm whether overexpression of the *SOD* and *POD* genes could increase SOD and POD activities. We analyzed 3 *ThSOD genes* (GenBank numbers: KF756930–KF756932) and 3 *ThPOD* genes (GenBank numbers: KF756934–KF756936). The results indicated that compared to that in control *T. hispida*, the OE plants indicated markedly increased expression of all the analysed *PODs* and *SOD*s in osmotic and salt stress environment, while their expression were markedly reduced in the IE plants (Figure 6). Three Arabidopsis *POD*s (AT2G18140, AT5G58400 and AT1G14550) and three Arabidopsis *SOD* genes (AT2G28190, AT3G10920 and AT1G08830) were assessed to test the expression patterns of them in Arabidopsis seedlings. Compared with the expression levels of the Col-0 plants, the *SODs* and *POD*s of all the studies were markedly advanced in transgenic lines in both salt and osmotic stress environments (Appendix A). The results showed that the expression of *SODs* and *PODs* could be induced by *ThNAC7*, which would result in enhanced SOD and POD enzyme activities.

### 2.8. Overexpression of ThNAC7 Reduces Water Loss Rate, Cell Death and Malondialdehyde Contents

Electrolyte leakage is associated with plant stress response, and stress-induced electrolyte leakage is often associated with the accumulation of ROS, usually leading to PCD (programmed cell death) [28]. The electrolyte leakage rates were measured to monitor cell death in *T. hispida* under different environment. They had not substantial difference in electrolyte leakage rates of the OE, IE, and control *T. hispida* plants in normal growth conditions, while the electrolyte leakage rates showed marked differences under osmotic and salt stress environment. The highest electrolyte leakage rates appeared in the IE plants, then in the control, and the lowest in OE plants (Figure 7A). The water loss assay indicated that the water loss rates of OE plants were lower than those in the control, while that of IE plants were the highest (Figure 7B). Moreover, the MDA contents were similar among the control, OE, and IE *T. hispida* plants in normal growth environment. Nevertheless, the highest MDA level was appeared in the IE plants, followed by the control plants, and the lowest levels were recorded in the OE *T. hispida* plants in osmotic and salt stress (Figure 7C). In addition, the electrolyte leakage rate, MDA content and water loss rate in transgenic *Arabidopsis* seedlings were all markedly decreased compared to those in Col-0 *Arabidopsis* plants under osmotic and salt stress (Appendix A). The results were richly consistent with the results from the *T. hispida* seedlings, proving that overexpression of *ThNAC7* could decrease the water loss rate, cell death, and MDA content in osmotic and salt stress environments to protect plants.

### 2.9. ThNAC7 Positively Affects Proline Biosynthesis

We next studied whether *ThNAC7* is involved in proline biosynthesis in salt or osmotic stress environment. The proline contents were measured in comparison of the control, OE, and IE *T. hispida* seedlings. There was no difference in proline contents among the above three *T. hispida* seedlings in normal environment; whereas, there were substantial differences in the proline contents among the OE, IE, and control *T. hispida* plants in osmotic or salt stress. The order of the proline contents from highest to lowest was the OE plants, the control plants, and the IE *T. hispida* plants (Figure 8A). We further studied whether the altered proline contents were caused by changes in expression of proline biosynthesis genes, including *ThP5CS* genes (delta 1-pyrroline-5-carboxylate synthetase, GenBank number: KM101096 and KM101097). There was no difference in expression of *ThP5CS1 and ThP5CS2* among all 3 types of plants in normal conditions. Under osmotic or salt stress, overexpression of *ThNAC7* induced the transcription levels of proline biosynthesis related genes, including *ThP5CS1* and *ThP5CS2* (Figure 8B,C).

To confirm the role of *ThNAC7* in proline biosynthesis, we repeated the above analyses on *Arabidopsis* transformed with Col-0 and *ThNAC7Arabidopsis* plants. In normal growth environment, the Col-0 and OE plants had similar proline contents. By contrast, under salt or osmotic stress, the proline levels in OE2 and OE5 lines were markedly increased compared with the Col-0 (Appendix A). We further studied the gene expression related to proline biosynthesis, i.e., *AtP5CS1* (AT2G39800) and *AtP5CS2* (AT3G55610). Under the salt or osmotic stress conditions, both *AtP5CS1* and *AtP5CS2* showed markedly higher expression in OE2 and OE5 lines than in Col-0 plants (Appendix A). These outcomes are in accordance with the results in *T. hispida*.

### 2.10. Identification of the Genes Regulated by ThNAC7 Using RNA-Seq

The *ThNAC7*-overexpressing *Arabidopsis* plants displayed higher tolerance than Col-0 *Arabidopsis* plants in salt stress; therefore, we performed an RNA-seq analysis of OE5 and Col-0 to verify the down-stream target genes of *ThNAC7* responsible for enhancing salt tolerance.

Under normal conditions, between OE5 and Col-0, a total of 272 differentially expressed genes (DEGs, log2 ratio >1, FDR < 0.05) were obtained, and 185 were upregulated and 87 were downregulated (Appendix A). Total of 673 DEGs were obtained, including 453 that were induced by *ThNAC7* and 220 that were inhibited by *ThNAC7* under salt stress (Appendix A). Among the DEGs upregulated under salt stress conditions with functional information, we found that over 40 encoded proteins and multiple TF genes that respond to salt stress (Table 1), such as peroxidase; lipid transfer protein; MAP kinase; late embryogenesis abundant protein; aquaporin; ethylene-responsive TF genes; and *MYB*, *HSF*, *DREB*, *TCP*, *NAC*, *bZIP*, and *WRKY* TF genes.

GO (Gene ontology) analyses showed that these DEGs were primarily related to 1 molecular function, 12 biological processes and three cellular components. Most were categorized in “metabolic process”, “single-organism process”, “cell part”, “cell”, and “organelle”. The other genes were assigned to “cell process”, “response to stimulus”, “biological regulation”, “cellular component organization or biogenesis”, “localization”, “development process”, “signaling”, “reproductive process”, “multicellular organismal process”, “reproduction”, “binding”, and “catalytic activity” (Appendix A).

Furthermore, we randomly selected 12 DEGs related to salt stress (six upregulated and six downregulated) and compared their expression levels between Col-0 and OE5 *Arabidopsis* plants by qRT-PCR (Table 2 and Appendix A). As shown in Figure 9 and Appendix A, all these genes had similar expression patterns to those decided using RNA-seq, proving the reliability of the RNA-seq results.

## 3. Discussion

Previously, we identified 16 NAC TFs that are involved in the differential response to NaHCO_3_ stress [27]. Furthermore, we confirmed that *ThNAC7* was greatly induced by salt stress; therefore, we selected *ThNAC7* for further research. Transactivation assays indicated that ThNAC7 has transcriptional activation that lies within three independent activation domains: aa 216–290, 291–360, and 361–435, respectively (Figure 2B). The full-length ThNAC7 protein showed strong transcriptional activation activity without the N-terminal domain (dc1, 1–435), containing three activation domains, abolished the transcriptional activation activity. These results indicated that ThNAC7 might possess a transcriptional repression domain in N-terminal 1–145 amino acids. The transactivation activity of NAC TFs might depend upon the interaction between the NAC activation domain and repression domain [29]. Therefore, whether ThNAC7 is a transcriptional activator or repressor depends on its interactions with other TFs or target genes.

The expression of *ThNAC7* was greatly induced by osmotic or salt stress conditions in *T. hispida* (Figure 3), indicating that *ThNAC7* plays roles in abiotic stress responses. To analyze the molecular function of *ThNAC7* in resistance to abiotic stress, two independent *ThNAC7* transgenic lines (OE2 and OE5) lines were used in further study. The overexpression of *ThNAC7* in transgenic *Arabidopsis* improved root growth (Figure 4A,B), fresh weight (Figure 4C), and chlorophyll contents (Figure 4D). Therefore, *ThNAC7* plays a positive role in osmotic or salt stress tolerance in plants.

ROS scavenging is crucial in resistance to plant abiotic stress [30,31,32,33]. We found that overexpression of *ThNAC7* could reduce excess ROS accumulation in this study (Figure 5 and Appendix A), indicating that *ThNAC7* is related to ROS scavenging. In addition, *ThNAC7* can induce the transcription levels of *PODs* and *SOD*s in transiently transformed *T. hispida* and stably transformed *Arabidopsis* (Figure 6 and Appendix A), thereby improving their activities (Figure 6 and Appendix A). Additionally, overexpression of *ThNAC7* could decrease cell death, water loss rate, and the MDA content in osmotic or salt stress (Figure 7 and Appendix A). These data showed that *ThNAC7* could elevate the transcription levels of ROS scavenging genes for improving stress tolerance. NAC TFs could enhance stress tolerance and modulate the expression level of ROS scavenging genes, which is accordance with previous studies [34,35,36,37]. Overexpression of *ThNAC13* enhanced the osmotic and salt stress tolerances via controlling the transcriptional levels of *PODs* and *SODs* to enhance the ROS scavenging ability [34]. In this study, we found that both ThNAC7 and ThNAC13 belong to NAC subfamily I in *T. hispida* (Appendix A).

Proline is not only an osmotic agent, but also a radical scavenger. It can maintain photosynthetic activity and defend cells from harm in abiotic stress environments, allowing plants to maintain stable growing conditions under long term stress [38,39]. We found that the transcription levels of *P5CS*s were induced by *ThNAC7* in *T. hispida* (Figure 8) and *Arabidopsis* (Appendix A). In addition, the transcription levels of *P5CSs* were positively related to the proline content (Figure 8). This result demonstrated that *ThNAC7* activates proline biosynthesis to increase the proline content, which consequently increases abiotic stress tolerance by elevating the osmotic potential and ROS scavenging ability.

The RNA-seq analyses revealed that more than 40 upregulated genes are related to salt stress, including four aquaporins (AQPs) and three lipid transfer proteins LTPs (Table 1). Many AQPs in the PIP subfamilies affect the expression levels of mRNA transcripts or/and proteins under salt, drought, or cold stresses [40]. In transgenic *Arabidopsis*, the overexpression of *OsPIP1* and *OsPIP2* improved drought and salt tolerance [41]. LTPs are responsive to many abiotic stresses [42,43,44]. These results suggested that *ThNAC7* might improve abiotic stress tolerance via inducing the transcription levels of *AQPs* and *LTPs*.

In addition, *ThNAC7* regulated multiple TFs involved in abiotic stress. Seven *ERF*s were up-regulated by *ThNAC7* in salt conditions. For instance, *RAP2.6* can response to high cold, salt, osmotic and ABA stresses [45]. Overexpression of *RAP2.6L* in *Arabidopsis* had no impact on phenotype, but it could improve drought and salt stresses tolerance [46]. The *ABR1* gene could response to drought, salt, cold and ABA stresses [47]. Therefore, the ERF TFs mediated by *ThNAC7* might play a vital role in salt stress tolerance. Moreover, other TF families were differentially expressed, such as MYB, bHLH, HSP, DREB, TCP, NAC, bZIP, and WRKY. *Arabidopsis* DREB2C plays vital roles in salt stress tolerance and the regulating of several genes related to abiotic stress [48]. The co-expression of *AtWRKY28* and *AtbHLH17* enhances resistance to NaCl, Mannitol, and oxidative stress in *Arabidopsis* [49]. *ATNAC2* and *ANAC092* are highly affected by salinity and can positively regulate the leaf senescence [50]. These results suggested that *ThNAC7* may directly or indirectly regulate these stress-related TFs to improve salt stress resistance in plants.

## 4. Materials and Methods

### 4.1. Plant Materials and Growth Conditions

Seeds of *T. hispida* were planted in containers with a mixture of turf peat and sand (3:1 *v*/*v*), were grown in the greenhouse maintained at 24 °C, with 70–75% relative humidity and a 14 h light/10 h dark photocycle. Seeds of *Arabidopsis thaliana* Columbia (Col-0) were directly seeded into containers with a mixture of perlite and soil (2:1 *v*/*v*) in the greenhouse at 22 °C, with 65–70% relative humidity and a 16 h light/8 h dark photocycle.

### 4.2. Plasmid Construction and Plant Transformation

In our previous study, eight transcriptomes were obtained from *T. hispida* [27]. The open reading frame (ORF) sequence of *ThNAC7* (GenBank number: JQ974961) was obtained using the transcriptome data. Multiple sequence alignments of ThNAC7 and 10 NAC proteins from different species were performed using ClustalW. The NAC proteins of *Arabidopsis* were obtained from UniPort (http://www.uniport.org/). Phylogenetic tree analysis was generated between ThNAC7 and 105 NAC proteins from *Arabidopsis* using the MEGA6.06 program (Tokyo Metropolitan University and 1 Research Center for Genomics and Bioinformatics, Tokyo, Japan). Another phylogenetic tree was generated using MEGA6.06 between ThNAC7 and 88 NAC proteins from *T. hispida.*

The ORF of *ThNAC7* was inserted into plasmid pROKII following the CaMV 35S promoter. The inverted repeat cDNA sequence of *ThNAC7* was cloned to the pFGC5941 RNAi vector (pFGC::ThNAC7) to knockdown the expression of *ThNAC7*. All the primers used in the experiments are listed in Appendix A. The highly efficient transient transformation of *T. hispida* plants were performed as previously described by Ji et al. [51]. We constructed three kinds of transgenic *T. hispida* plants: OE (overexpression) plants (transformed with 35S::ThNAC7), IE (inhibited expression) plants (transformed with pFGC::ThNAC7), and control *T. hispida* plants with the empty pROKII plasmid. The 35S::ThNAC7 plasmid was stably transformed into *Arabidopsis* by the Agrobacterium mediated floral dip transformation method to obtain ThNAC7 overexpressed plants [52]. T_3_ homozygous transgenic *Arabidopsis* were used in further study.

### 4.3. Subcellular Location of ThNAC7 Proteins

The coding sequence (CDS) of *ThNAC7* with no termination codon was fused with the C-terminus of the *GFP* following the CaMV 35S promoter. 35S::GFP was used as control. All primer used are listed in Appendix A. The 35S::ThNAC7-GFP and 35S::GFP plasmids was respectively imported into onion epidermal cells by particle bombardment (Bio-Rad, Hercules, CA, USA). The nuclei were stained using 4′,6-diamidino-2-phenylindole (DAPI) (10 μg·mL^−1^) with phosphate buffered saline for 5 min. The transformed onion epidermal cells were visualised by LSM700 confocal laser scanning microscopy (Zeiss, Jena, Germany).

### 4.4. Transactivational Activity of ThNAC7

To evaluate the transcriptional activity of *ThNAC7* and determine its transactivation domain, the full length or various truncated *ThNAC7* fragments were fused to the yeast *GAL4* DNA binding domain of the pGBKT7 (Clontech, Palo Alto, CA, USA) using Infusion enzyme (Clontech) and transformed in Y2HGold cells. The procedure was performed according to the Matchmaker™ Gold Yeast Two-Hybrid System instructions. In the transactivation assay, the transformed yeast cells were grown on SD/-Trp-His/X-α-gal medium for 3–5 days. All primers were listed in Appendix A. Three independent experiments were performed.

### 4.5. Stress Tolerance Assays

The T_3_ generation *ThNAC7* transgenic *Arabidopsis* lines were used as the materials for stress tolerance analyses. After culturing in 1/2 MS medium for 5 days, the germinated *Arabidopsis* seeds were separately transferred to 1/2 MS or 1/2 MS containing 150 mM mannitol or 150 mM NaCl for two weeks. The fresh weight and root length of *Arabidopsis* were determined. The chlorophyll contents of excised leaves were measured as described by Lichtenthaler [53]. *Arabidopsis* seeds cultured in 1/2 MS plates for five days were planted in soil. The three-week seedlings were watered using 400 mM mannitol or 250 mM NaCl for 2 weeks; the fresh weights of plants were also determined. Three independent experiments were performed.

### 4.6. Physiological Measurements

*T. hispida* seedlings were cultured in 1/2 MS or 1/2 MS with 200 mM mannitol or 150 mM NaCl for 24 hours were for physiological assays. *Arabidopsis* grown into soil for 3 weeks were watered with 200 mM mannitol or 150 mM salt (NaCl) solution at 24 h for physiological assay. The seedlings were treated with 200 mM mannitol or 150 mM NaCl solution for 2 hours. The leaves of *T. hispida* or *Arabidopsis* plants was stained by nitroblue tetrazolium (NBT) and diaminobenzidine (DAB) using the methods of Fryer et al. [54,55]. The SOD and POD activities, and the MDA content were determined using the methods of Wang et al. [56]. The H_2_O_2_ level was detected according to the method of Dal Santo et al. [57]. An electrolyte leakage assay was determined via the method of Ji et al. [51]. The water loss rate was measured using the method of Zhang et al. [55]. The proline content was determined according to the method of Bates et al. [58]. Three independent experiments were performed.

### 4.7. qRT-PCR Assay

The total RNA of *T. hispida* was obtained using the Universal Plant Total RNA Extraction Kit (BioTeke). *Arabidopsis* total RNA was isolated using the Trizol reagent (Invitrogen, Waltham, MA, USA). Then RNA (2 μg) was reversely transcribed into cDNA as PCR templates. *Actin* (GenBank number: FJ618517) and *β-tubulin* (GenBank number: FJ618519) were used as reference genes for *T. hispida* analysis (Appendix A). For *Arabidopsis* analysis, *α-tubulin* (AT1G50010) and *Actin3* (AT3G53750) were used as reference genes (Appendix A). The reaction system included 2 μL of cDNA template, 0.5 μM each of primer, and 10 μL of SYBR Green Real time PCR Master Mix (Toyobo) in the volume of 20 μL. The procedure involved the following: 94 °C/30 s; followed by 45 cycles at 94 °C/12 s, 60 °C/30 s, and 72 °C/40 s; followed by 1 s at 82 °C for plate reading. The qRT-PCR was performed using an Opticon 2 System (Bio-Rad, Hercules, CA, USA). For each sample, the melting curve evaluated the purity of the PCR products. The expression levels of genes were treated using the 2^−ΔΔ*C*t^ method [59]. Three independent experiments were performed.

### 4.8. RNA-Sequencing Analysis

Total RNA from four-week-old *ThNAC7*-overexpression line OE5 and Col-0 *Arabidopsis*, without treatment or treated with NaCl (150 mM) for 6 hours, were used for RNA-sequencing, and three biological replicates were performed. Total RNAs were extracted using the Trizol reagent, and purified using AMPure beads (Agencourt, CA, USA). The preparation of the sequencing library and PCR amplification used a TruSeq PE Cluster Kit and a TruSeq™ DNA Sample Prep Kit-Set A, respectively (Illumina, San Diego, CA, USA). RNA-seq was analyzed on the Illumina HiSeq 2000 platform. The relative expression levels were obtained under salt and normal conditions. All the relative expression levels were calculated using log_2_. With a log-fold expression change of |log_2_FC| ≥ 1, and the threshold of false discovery rate (FDR < 0.05), DEGs were identified via the DEGSeq algorithm [60]. The correlation between qRT-PCR and digital genes expression (DGE) was evaluated statistically by calculating the Pearson correlation coefficient. Gene ontology (GO) classification was executed using the molecule annotation system (MAS). The RNA-seq data was uploaded to the NCBI (PRJNA525176).

### 4.9. Statistical Analyses

Statistical analyses were accomplished with SPSS (IBM SPSS 22, IBM Corp., Armonk, NY, USA). Data were calculated using student’s *t*-test. *p* < 0.05 was statistically significant. * represents *p* < 0.05 in the figures. Three biological replicates were carried out for statistical analyses.

## 5. Conclusions

A NAC TF from *Tamarix hispida*, ThNAC7, is a nuclear localization protein with transactivation activity. Overexpression of *ThNAC7* markedly increased proline levels and enhanced the POD and SOD activities to increase the ROS scavenging ability in both transiently transformed *T. hispida* and stably transformed *Arabidopsis*. *ThNAC7* positively regulated a series of genes that enhance osmotic and salt stress tolerance, including salt tolerance genes and many TF genes. Taken together, *ThNAC7* promotes the expression level of genes related to stress tolerance in order to enhance osmotic and salt stress tolerance by improving the osmotic potential and increasing ROS scavenging. The results of the present study enrich the molecular mechanism of *ThNAC7* involved in salt and osmotic stresses.

## Figures and Tables

**Figure 1 plants-08-00221-f001:**
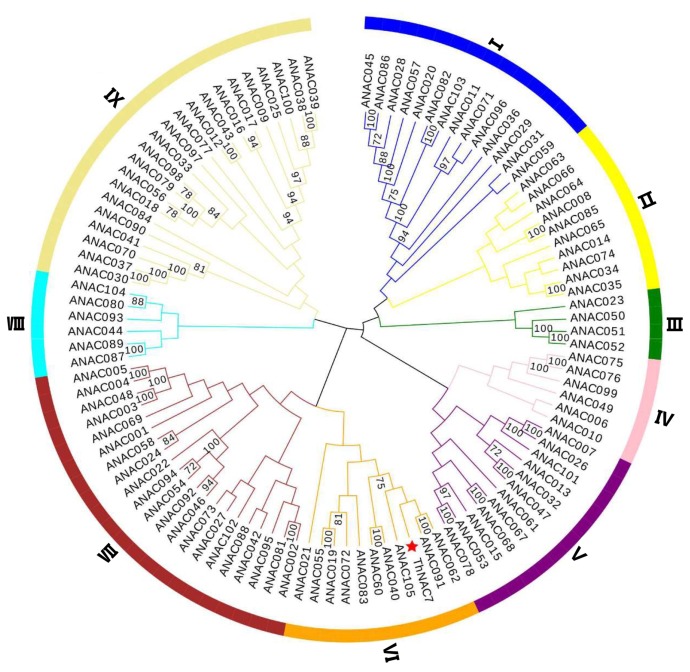
Phylogenetic analysis between ThNAC7 and the NAC proteins from *Arabidopsis*. The phylogenetic relationship of ThNAC7 and the NAC proteins from *Arabidopsis*. The ThNAC7 and 105 *Arabidopsis* NACs were aligned; the unrooted NJ tree was constructed using MEGA 6.06.

**Figure 2 plants-08-00221-f002:**
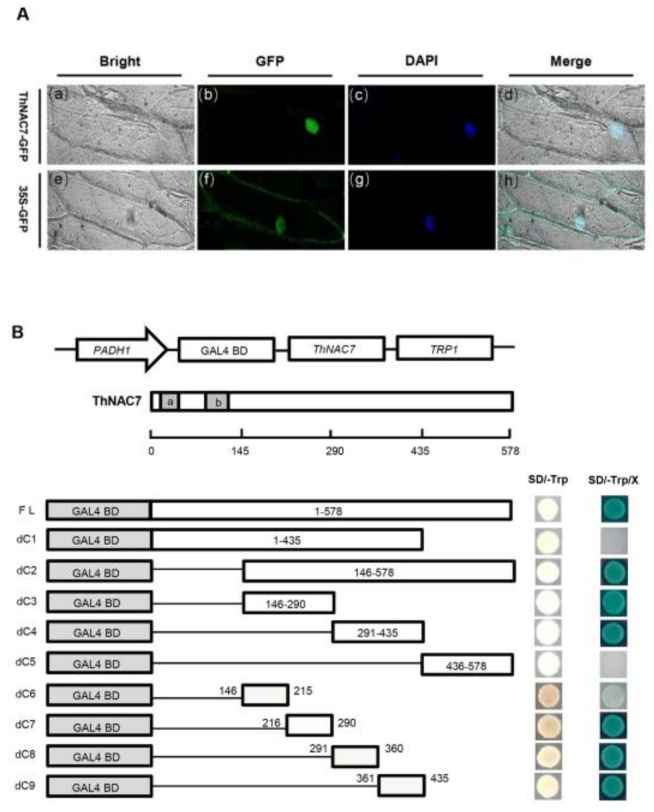
Subcellular localization and transcriptional activation of *ThNAC7*. (**A**): Subcellular localization analysis of *ThNAC7*. The onion nuclei were visualised through DAPI (4′,6-diamidino-2-phenylindole) staining. Bars, 50 μm. (**B**): Transactivation assay of ThNAC7 (a, b: NAC domain). A diagram of the pGBKT7 constructs for expressing different truncated ThNAC7 proteins in yeast cells. Transactivation assay of the intact or truncated ThNAC7 proteins. Full length or truncated CDSs of *ThNAC7* were constructed into pGBKT7 vector and transformed into Y2HGold cells, and grown on SD/-Trp or SD/-Trp/-His/X-α-gal mediums to assess their transcriptional activation.

**Figure 3 plants-08-00221-f003:**
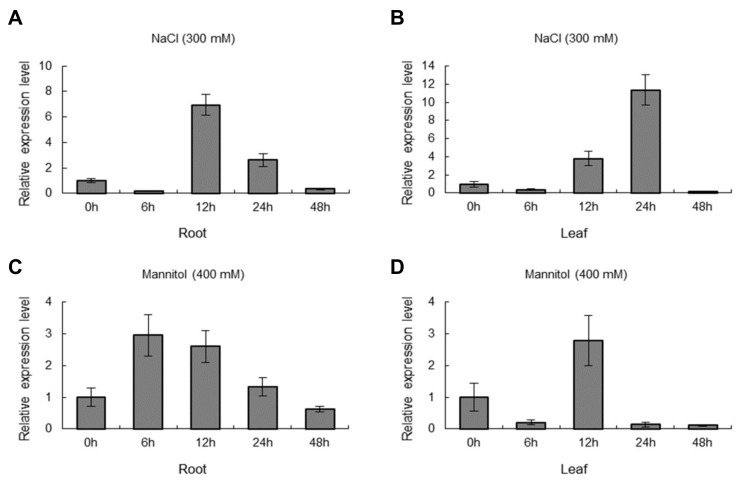
The expression level of *ThNAC7* in response to osmotic or salt stress in *T. hispida*. The expression patterns of *ThNAC7* in roots or leaves of *T. hispida* plants in response to treatment with NaCl (300 mM) (**A**,**B**) or mannitol (400 mM) (**C**,**D**). The expression of *ThNAC7* in normal conditions (0 h) was designed as 1 to standardize the expression level of *ThNAC7* in salt or osmotic conditions.

**Figure 4 plants-08-00221-f004:**
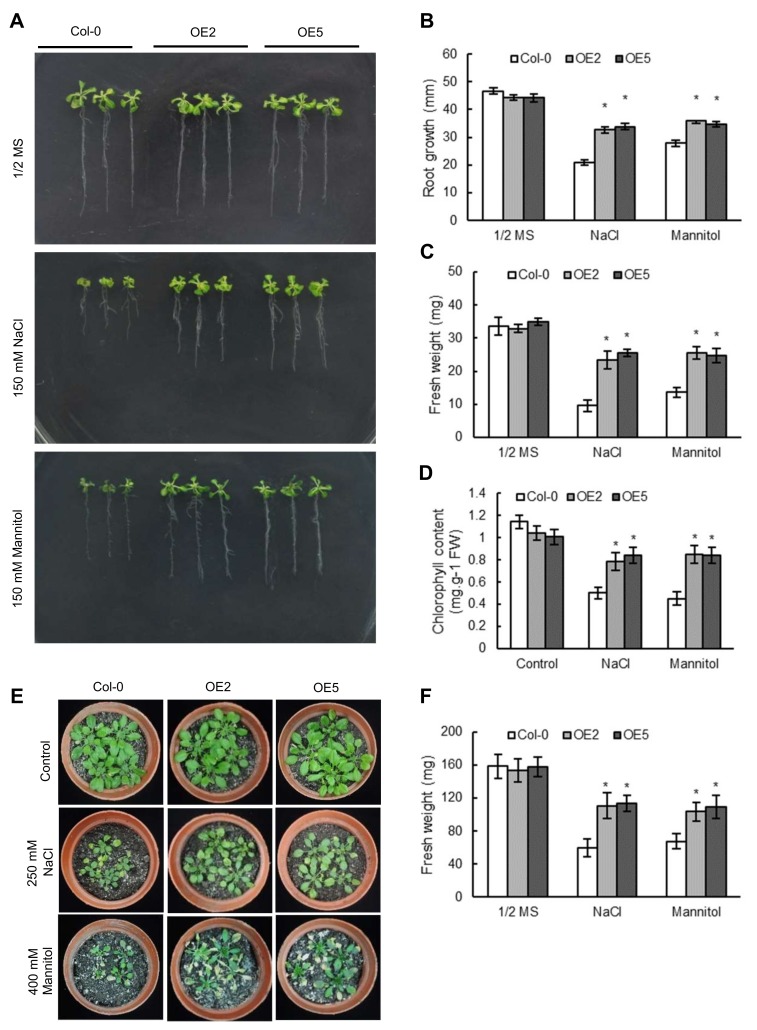
Stress tolerance of overexpressing *ThNAC7* in *Arabidopsis*. (**A**): Growth phenotype of transgenic and Col-0 plants under normal (1/2 MS), salt (150 mM NaCl), or osmotic (150 mM mannitol) stress. (**B**): Root length assay. (**C**): Analysis of fresh weight. (**D**): Chlorophyll content assay. The plants were cultured in 1/2 MS (as control) or 1/2 MS containing 150 mM NaCl or 150 mM mannitol for 14 days for the above analysis. (**E**): Growth phenotype of *Arabidopsis* plants. (**F**): Fresh weight of *Arabidopsis* seedlings. Three weeks old *Arabidopsis* seedlings in soil were treated with 400 mM mannitol or 250 mM NaCl for 2 weeks, then took a photograph of their phenotypes. Control: Plants growing in normal conditions. The asterisks (*p* < 0.05) show * significant differences compared to Col-0 *Arabidopsis*. Col-0: wild-type *Arabidopsis* plants; OE2 and 5: *Arabidopsis* lines 2 and 5 transformed with *ThNAC7*.

**Figure 5 plants-08-00221-f005:**
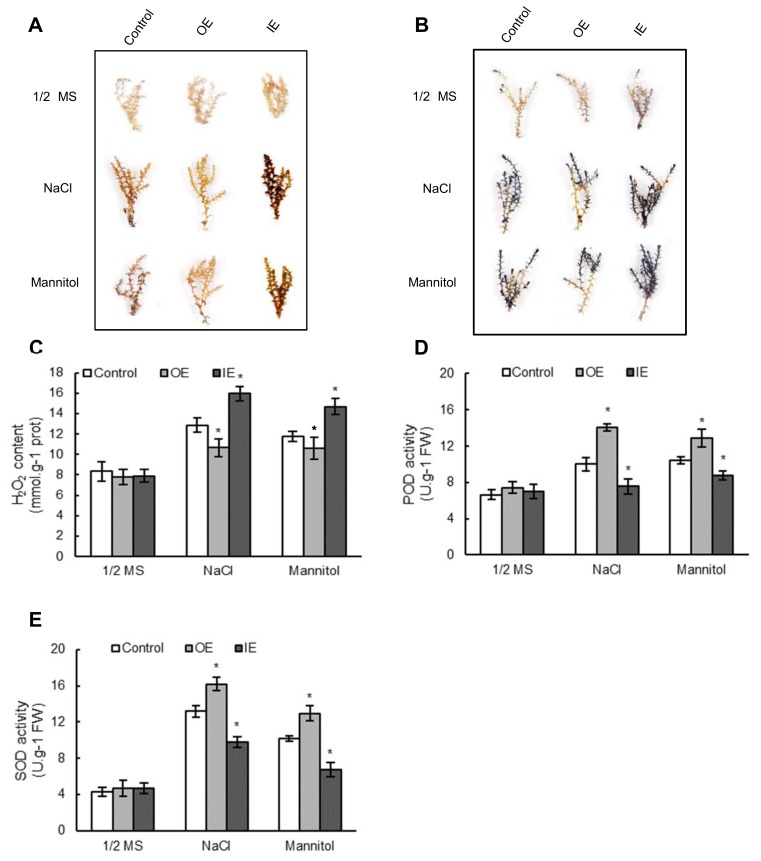
Analysis of ROS accumulation and the activities of SOD and POD among OE, IE, and control *T. hispida* plants. (**A**,**B**): *T. hispida* plants treated with NaCl (150 Mm) or mannitol (200 mM), and stained with DAB to visualise H_2_O_2_ level (**A**), or stained with NBT to visualise O_2_^−^ (**B**). (**C**): Measurement of H_2_O_2_ levels in *T. hispida*. (**D**,**E**): Measurement of the POD (**D**) and SOD (**E**) activity. OE: *T. hispida* plants overexpressing *ThNAC7*; IE: *ThNAC7* RNAi-silenced *T. hispida* seedlings; Control: *T. hispida* plants transformed with pROKII.

**Figure 6 plants-08-00221-f006:**
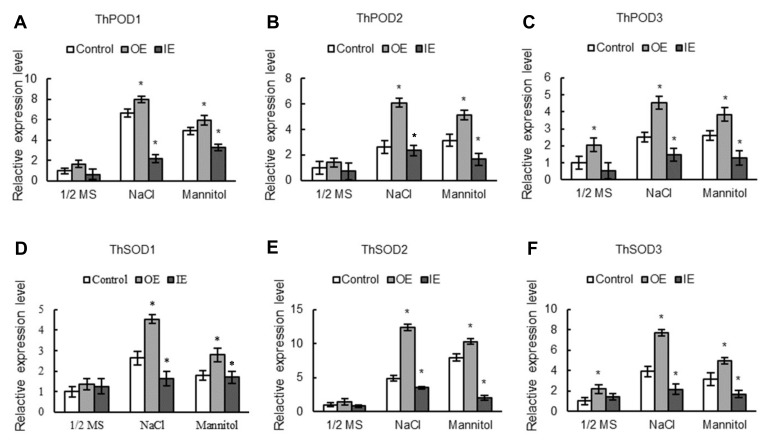
The expression patterns of the *PODs* and *SODs* in OE, IE, and control *T. hispida* seedlings. The expression of *PODs* (**A**) and *SODs* (**B**) were analyzed in control, OE, and IE *T. hispida* in response to salt (150 mM NaCl) or osmotic (200 mM mannitol) stress. The transcription level of a gene in control under normal growth conditions was designed as 1 to standardize its relative expression values. OE: *T. hispida* seedlings overexpressed *ThNAC7*; IE: *ThNAC7* RNAi-silenced *T. hispida* seedlings; Control: *T. hispida* seedlings transformed with pROKII. * Significant (*p* < 0.05) difference compared to the control plants.

**Figure 7 plants-08-00221-f007:**
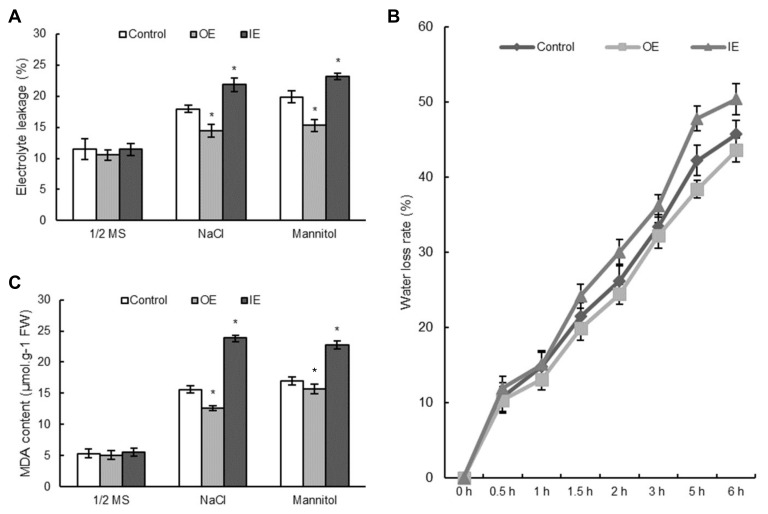
Detection of cell death, water loss rates, and MDA level in OE, IE, and control *T. hispida* seedlings. (**A**): Electrolyte leakage rates assay. (**B**): Water loss rates. (**C**): MDA content. The *T. hispida* plants were treated with H_2_O (as control), NaCl (150 mM), or mannitol (200 mM) for 24 h. Data are means ± SD from 3 independent experiments. * Significant (*p* < 0.05) difference compared to the control. OE: *T. hispida* overexpressing *ThNAC7*; IE: *ThNAC7* RNAi-silenced *T. hispida*; Control: *T. hispida* plants transformed with pROKII.

**Figure 8 plants-08-00221-f008:**
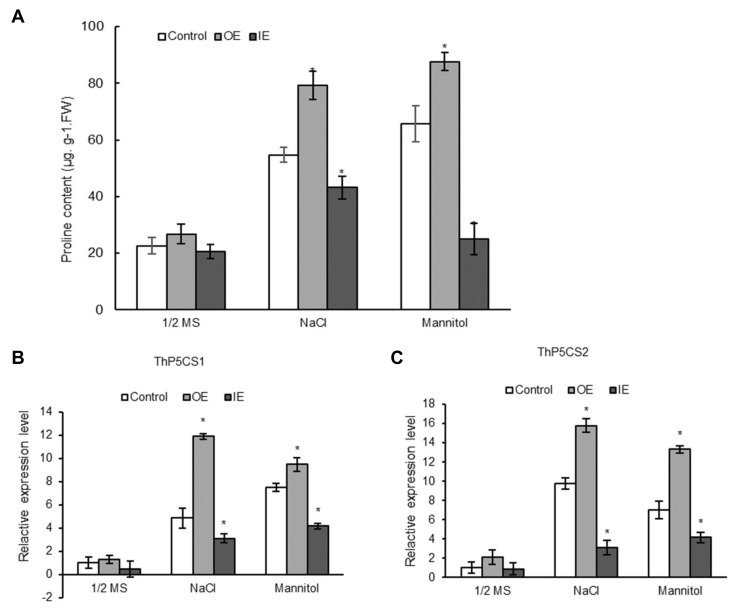
Analysis of the effect of *ThNAC7* on the biosynthesis of proline. (**A**): Analysis of proline contents in OE, IE, and control *T. hispida* seedlings in response to osmotic (200 mM Mannitol) or salt (150 mM NaCl)stresses. (**B**,**C**): The expression levels of proline biosynthesis related genes. The relative expression levels were log2 transformed. Asterisk indicate significant (*p* < 0.05) difference compared to the control. OE: *T. hispida* plants overexpressing *ThNAC7*; IE: *ThNAC7* RNAi-silenced *T. hispida* seedlings; Control: *T. hispida* plants transformed with pROKII.

**Figure 9 plants-08-00221-f009:**
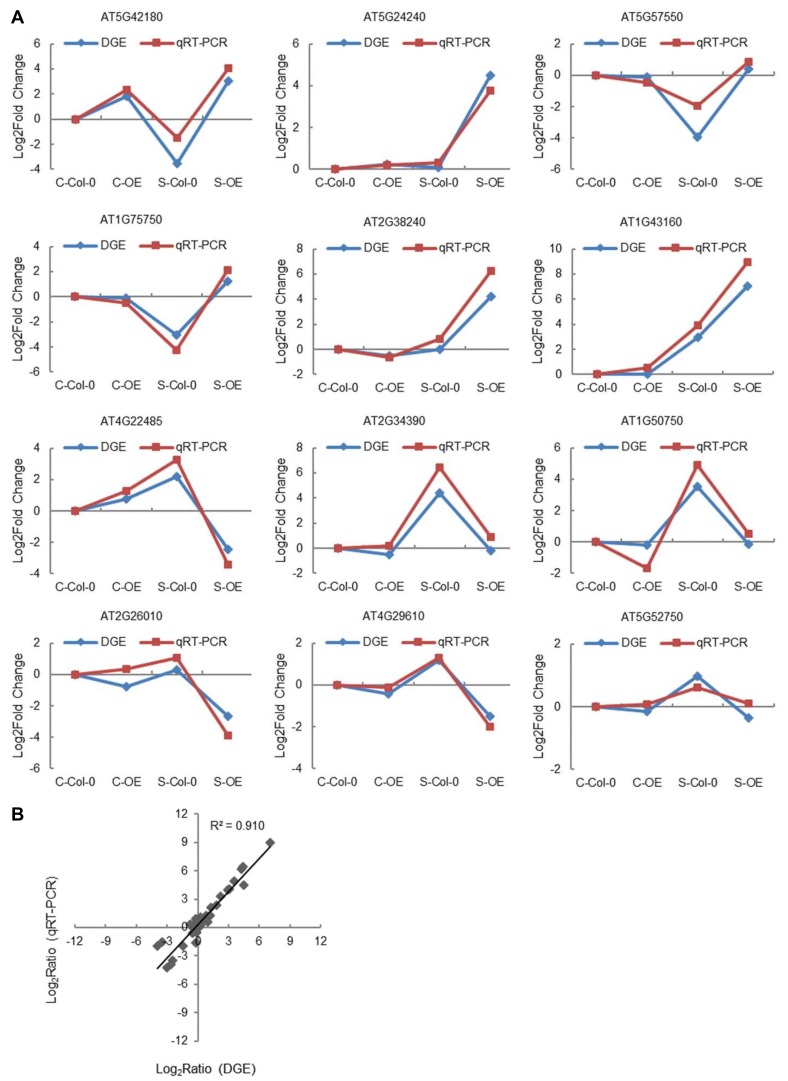
Comparison of the expression patterns between RNA-seq and qRT-PCR. **A**: Twelve highly differentially regulated genes were randomly selected for qRT-PCR assay. **B**: Correlation: correlation analysis of the results between qRT-PCR and RNA-sequencing; the correlation coefficient R^2^ is 0.910. * Significant (*p* < 0.05) difference compared with the control plants.

**Table 1 plants-08-00221-t001:** List of DEGs upregulated in response to salt stress in *ThNAC7*-overexpression line OE5 compared with Col-0 *Arabidopsis* plants.

Locus ID	Gene Name	FDR	Fold Change	Annotations-Description
AT5G42180	PER64	2.62 × 10^−8^	6.65	peroxidase
AT1G43160	RAP2.6	6.88 × 10^−16^	4.12	ethylene-responsive transcription factor RAP2-6
AT1G09540	MYB61	1.02 × 10^−6^	3.44	myb domain protein 61
AT5G64750	ABR1	8.68 × 10^−20^	3.04	ethylene-responsive transcription factor ABR1
AT2G45180	LTP	1.23 × 10^−21^	2.78	lipid transfer protein (LTP) family protein
AT2G32510	MAPKKK17	5.09 × 10^−9^	2.38	mitogen-activated protein kinase kinasekinase 17
AT1G52690	LEA	0.004977	2.31	Late embryogenesis abundant protein (LEA) family protein
AT1G21910	DREB26	2.03 × 10^−6^	2.24	ethylene-responsive transcription factor ERF012
AT4G29930	bHLH27	1.36 × 10^−7^	2.2	transcription factor bHLH27
AT1G05100	MAPKKK18	0.008566	2.18	mitogen-activated protein kinase kinasekinase 18
AT4G22470	LTP	6.33 × 10^−10^	2.12	lipid transfer protein (LTP) family protein
AT5G59310	LTP4	1.96 × 10^−7^	2.11	non-specific lipid-transfer protein 4
AT2G37180	RD28	6.49 × 10^−44^	2.1	aquaporin PIP2-3
AT5G12020	HSP17.6II	0.008487	2.1	class II heat shock protein 17.6
AT2G20880	ERF053	0.00016	2.09	ethylene-responsive transcription factor ERF053
AT2G40340	DREB2C	0.000646	2.04	dehydration-responsive element-binding protein 2C
AT5G61890	ERF114	0.001014	2.03	ethylene-responsive transcription factor ERF114
AT4G05100	AtMYB74	2.65 × 10^−7^	2.01	myb domain protein 74
AT3G18550	BRC1	0.000907	1.93	transcription factor TCP18
AT4G18170	WRKY28	2.97 × 10^−7^	1.75	DNA binding-like protein
AT5G13330	RAP2.6L	3.84 × 10^−12^	1.71	ethylene-responsive transcription factor ERF113
AT1G67260	TCP1	2.94 × 10^−7^	1.71	transcription factor TCP1
AT1G71030	ATMYBL2	2.28 × 10^−8^	1.64	putative myb family transcription factor
AT5G39610	ATNAC2	9.09 × 10^−6^	1.63	NAC-domain transcription factor
AT2G22200	ERF056	4.68 × 10^−5^	1.63	ethylene-responsive transcription factor ERF056
AT1G75490	DREB2D	0.001527	1.6	dehydration-responsive element-binding protein 2D
AT4G23400	PIP1D	3.65 × 10^−11^	1.57	putative aquaporin PIP1-5
AT5G40630	-	0.001874	1.56	ubiquitin family protein
AT2G22770	bHLH020	6.89 × 10^−6^	1.44	putative bHLH transcription factor
AT2G26150	ATHSFA2	2.19 × 10^−6^	1.43	heat stress transcription factor A-2
AT3G53420	PIP2A	1.95 × 10^−13^	1.4	aquaporin PIP2-1
AT5G56550	OXS3	1.04 × 10^−10^	1.37	protein OXIDATIVE STRESS 3
AT2G21650	MEE3	0.006908	1.36	MYB transcription factor RSM1
AT4G27410	RD26	6.73 × 10^−5^	1.33	NAC transcription factor RD26
AT5G60890	ATMYB34	1.14 × 10^−7^	1.31	myb domain protein 34
AT1G56650	PAP1	2.41 × 10^−5^	1.28	transcription factor MYB75
AT1G54100	ALDH7B4	1.61 × 10^−8^	1.28	aldehyde dehydrogenase 7B4
AT5G28770	ATbZIP63	1.08 × 10^−13^	1.27	basic leucine zipper 63
AT5G50915	bHLH137	3.16 × 10^−5^	1.21	transcription factor bHLH137
AT2G36830	TIP	2.04 × 10^-10^	1.21	aquaporin TIP1-1

**Table 2 plants-08-00221-t002:** The expression values of RNA-seq for validation of the selected DEGs.

Locus ID	FDR	Fold Change	Annotations-Description
AT5G42180	2.62 × 10^−8^	6.65	peroxidase
AT5G24240	0.01	4.44	phosphatidylinositol 3- and 4-kinase/ubiquitin family protein
AT5G57550	3.65 × 10^−20^	4.35	probable xyloglucan endotransglucosylase/hydrolase protein 25
AT1G75750	3.37 × 10^−38^	4.26	GA-responsive GAST1 protein-like protein
AT2G38240	2.86 × 10^−13^	4.21	2-oxoglutarate (2OG) and Fe (II)-dependent oxygenase-like protein
AT1G43160	6.88 × 10^−16^	4.12	ethylene-responsive transcription factor RAP2-6
AT4G22485	5.48 × 10^−4^	−4.81	protease inhibitor/seed storage/LTP family protein
AT2G34390	6.8 × 10^−14^	−4.7	aquaporin NIP2-1
AT1G50750	5.5 × 10^−9^	−3.71	plant mobile domain family protein
AT2G26010	1.24 × 10^−5^	−3.07	plant defensin 1.3
AT4G29610	2.42 × 10^−14^	−2.73	cytidine/deoxycytidylate deaminase family protein
AT5G52750	2.14 × 10^−8^	−1.36	alpha/beta-hydrolases superfamily protein

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
