# Peer review of "The NAC Protein from Tamarix hispida, ThNAC7, Confers Salt and Osmotic Stress Tolerance by Increasing Reactive Oxygen Species Scavenging Capability"

_plants, 2019, doi:10.3390/plants8070221_

Round 1

Reviewer 1 Report

In this MS author explain about ThNAC7 gene from Tamarix hispida that confers salt and osmotic stress tolerance by increasing reactive oxygen species scavenging capability. overall MS is very well written. For the betterment of the MS I have few comments :

Major:

1. How you generated OX lines ? (Draw map of vector in the main figure 4. Which generation of seeds you used for stress test are they stable? how many copy number?

2. Introduction part is short add more references like:

a. 

Over-expression of dehydrin gene, OsDhn1, improves drought and salt stress tolerance through scavenging of reactive oxygen species in rice (Oryza sativa L.)

b. 

Overexpression of annexin gene AnnSp2, enhances drought and salt tolerance through modulation of ABA synthesis and scavenging ROS in tomato.

3. How many times you have repeated ROS experiment?

4. Please use better resolution images.

Reviewer 2 Report

The paper by He et al. addresses a topic of great interest in understanding osmotic and salt stress tolerance. This manuscript aims to characterize the NAC7 gene from Tamarix. The study attempts to understand the underlying mechanism of stress tolerance conferred by NAC7, which alterations in radicle scavenging activity. Furthermore, it appears that the experiments in this study are vividly performed to establish the roles of NAC7 their cross-talk in the modulation of salt and osmotic stress tolerance. The authors take several well-established, complementary approaches, including RNA-seq analyses, gene expression analyses, transactivation assays and metabolite measurements. I have no major concerns on this version of the manuscript. 

Minor comments:

1. The manuscript needs a through editing for English language and grammar. 

2. Throughout the methods, please indicate how many biological and technical replicates were involved in each of the experiments in their respective sub-sections.  

3. The RNA-seq experiments were conducted in Arabidopsis background. The authors may need to substantiate for the experimental design.

Reviewer 3 Report

The manuscript by He et al, in continuance of the groups previous work (Wang, L.; Li, Z.; Lu, M.; Wang, Y. ThNAC13, a NAC Transcription Factor from Tamarix hispida, Confers Salt and Osmotic Stress Tolerance to Transgenic Tamarix and Arabidopsis. Front. Plant Sci. 2017, 8, 635. ) describes a new NAC TF of Tamarix hipsida, ThNAC7, overexpression of which confers resistance to salinity and drought stresses, in both plant species Tamarix hipsida and Arabidopsis thaliana. The resistance is attributed to the limited production of ROS enhanced SOD and POD activity in the transgenic lines. The experimental design is solid and the manuscript is well written, but some amendments need to be performed before considering the manuscript for publication.

Major points

1)      Line 79: How did the authors choose NAC orthologs of other species? Are these the homologs of ThNAC7 or did they choose randomly? Please explain.

2)      Line 89 (Figs 1 and S1): Since Arabidopsis NACs are characterized and we know their classification, the authors could try to construct another phylogenetic tree using all Arabidopsis NACs plus the 88 ThNACs to see how they all classify together. Will they separate between the species or could the ThNACs blend with the Arabidopsis subgroups? This would be interesting.

3)      Figure 2A: The Image quality is very low. Please improve.

4)      Figure 3: Which is the reference condition that the treatments are compared with? The authors state “The expression level of ThNAC7 under normal growth conditions was designed as 1 to normalize the expression ThNAC7 under salt or osmotic conditions” but this is totally absent from both graphs. Please reconstruct the expression graphs properly, including the control conditions and state it clearly.

5)      Figures 4A and 4E: Provide higher resolution photographs. I do not know if it is due to the submission file or the original files, but better resolution photographs are necessary and would sure improve the manuscript.

6)      Figure 5A: Again, higher resolution images.

7)      Figure 9: Please indicate the 6 upregulated and 6 downregulated genes in a separate table with their RNA-seq expression values. Also, please add a supplemental figure of the q-RT-PCR charts performed for the genes presented. Figure 9 describes the trend of the 2 methods compared. I would also like to see the actual data of the two methodologies.

Minor points

1)      Line 22 (and throughout the manuscript): stably transformed

2)      Line 101: GFP is localized in the cytoplasm and it is also known that it passively diffuses into the nuclei, due to its’ small molecular weight. Please correct “the cell” with “the cytoplasm and nucleus”

3)      Line 110: It would helpful to provide a graphical cartoon of the ThNAC7 protein with the positions of their functional domains, so it would be easier for the reader to understand what deletions were made and which are the locations of the transcriptional activation domains.

4)      Figure 2B: On the second cartoon of ThNAC7 there are two gray boxes highlighted (a and b). There is no citation of what do they represent. Please mention on the legend.

5)      Line 152: weights à weight

6)      Line 153: had the approximately à had approximately the same…

7)      Line 155: Correct the citation on the Figure 4D.

8)      Line 157: Correct the citations. 4F à 4E and 4G à4F

9)      Line 191: Syntax.

Line 348 - 350: The authors state “indicating that ThNAC7 is likely to participate in the regulation of the chlorophyll synthesis pathway under salt and osmotic stress”. I think it is a strong statement to say and it is not well supported by the data presented. It could also just be a side effect due to the stress conditions. In fact, the authors, on the previous sentence (lines 345-346) state “Chlorophyll plays an important role in photosynthesis, and salt and drought stress can decrease the photosynthetic efficiency”, which I also believe is the case, so please scale it down a little.

Round 2

Reviewer 1 Report

I am Happy with the comments. This MS can be accepted in its current format.

Author Response

Dear reviewer, we are very grateful to your valuable suggestions.

Reviewer 3 Report

I am pleased with the changes and the response of the authors to my comments. I just have some more minor points that if addressed would only improve the manuscript before final acceptance.

1)      Line 309: impacted -- > introduced

2)      Line 358: treating -- > treatment

3)      Line 424: add osmotic (Mannitol) stress conditions

4)      Line 491: Figures S5A, B, E. The authors should change the succession of the figure panels so the citing would be hierarchical (E becomes C, and C and D become D and E, respectively). In fact, in Figure 5, this is the series they have kept (ROS stain photos, H2O2 content, POD and SOD activity).

5)      Line 500: Figures S5C, D (D and E, after the change)

6)      Line 559: I assume that authors wanted to say transgenic lines instead of Col-0 plants. Also, in Figure S6, the authors could provide the gene name synonyms (wherever available) on the chart titles (eg. AtSOD1, 2 or whichever it is). Also, instead of A, B, … F they could just divide the Figure in A -- > POD genes and B -- > SOD genes. The latter could also be applied to Figure 6.

7)      Line 578: I think that the Figure 7A reference should be also added there at the end of the sentence.

8)      Line 1027: overexpressional -- > transgenic or plants overexpressing the construct, etc…..

9)      Line 1036: introduced is not appropriate there. Transformed would suit better.

10)   Line 1185: suffered from -- > treated with

11)   Line 1219: qRT-PCR detections -- > qRT-PCR analysis

Lines 1217 – 1226 (Supp Figure legends): The authors wherever they have used panel separations in their Figures (A, B, C, …) they should also include them in the legends.
